# Occurrence and Seasonal Monitoring of Domoic Acid in Three Shellfish Species from the Northern Adriatic Sea

**DOI:** 10.3390/toxins14010033

**Published:** 2022-01-03

**Authors:** Kristina Kvrgić, Tina Lešić, Natalija Džafić, Jelka Pleadin

**Affiliations:** 1Veterinary Centre Rijeka, Croatian Veterinary Institute, Podmurvice 29, 51000 Rijeka, Croatia; kvrgic.vzr@veinst.hr (K.K.); dzafic.vzr@veinst.hr (N.D.); 2Laboratory for Analytical Chemistry, Croatian Veterinary Institute, Savska Cesta 143, 10000 Zagreb, Croatia; lesic@veinst.hr

**Keywords:** domoic acid, seasonal monitoring, bivalves, ascidians

## Abstract

As filter feeders, bivalves and ascidians can accumulate contaminants present in the environment and pass them on to higher food chain levels as vectors. The consumption of bivalves contaminated with the potent neurotoxin domoic acid (DA) can cause amnesic shellfish poisoning in humans. The aim of this study was to determine seasonal differences in occurrence and accumulation of this phycotoxin in European oysters (*Ostrea edulis* Linnaeus, 1758) (*n* = 46), Queen scallops (*Aequipecten opercularis* Linnaeus, 1758) (*n* = 53), and edible ascidians of the *Microcosmus* spp. (*n* = 107), originating from the same harvesting area in the Northern Adriatic Sea. The quantification was performed using ultra-performance liquid chromatography–tandem mass spectrometry (LC-MS/MS) preceded by derivatization with dansyl chloride. DA was found in very low concentrations throughout the year, with a maximum value of 810 μg/kg in Queen scallops. This study reveals differences in the occurrence and accumulation of DA between Queen scallops and the other two investigated species (oysters and ascidians) and the highest concentrations during the colder part of the year. Even though DA was detected in all of them, Queen scallops showed higher DA accumulation compared to the other two (*p* < 0.001), hence representing a sentinel species suitable for the monitoring of DA level in seafood.

## 1. Introduction

Due to its high nutritional value and the absence of additives, the consumption of seafood is essential in preventing non-communicable diseases such as cardiovascular diseases, as well in combating undernutrition and micronutrient deficiencies. Owing to their low amount of fat and a favourable fatty acid profile, as well as to their high share of easily digestible proteins containing essential amino acids, vitamins, and minerals, bivalves play an important role in a healthy and balanced diet [1,2,3,4]. Edible tunicate species such as ascidians are marine animals that represent an easily accessible source of valuable nutrients and bioactive compounds [5]. However, as filter feeders, bivalves and ascidians can accumulate contaminants present in the environment and pass them on to higher food chain levels as vectors [6].

The consumption of seafood contaminated with the potent neurotoxin domoic acid (DA) can cause severe amnesic shellfish poisoning (ASP) of humans with a possibly lethal outcome [7]. DA is a heat-stable, water-soluble tricarboxylic amino acid with a secondary amino group [8,9]. It is produced by marine diatom species of the *Pseudo-nitzschia* genus, red algae termed *Chondria armata*, and some other species belonging to the *Rhodomelaecae* family [9]. DA toxicity arises from its structural similarity with glutamic acid. As a glutamate agonist, DA binds to the kainate receptors in vital organs, causing neuronal depolarization [10]. The symptoms of intoxication are gastrointestinal, cardiovascular, and neurological in nature, whereas their intensity thereby depends on the ingested DA dose and individual susceptibility [9]. Clinical symptoms of ASP include cramps, vomiting, diarrhoea, headache, disorientation, and a characteristic short-term memory loss, while severe cases present with seizures, coma, and death [8]. The elderly and people with diabetes, kidney disorder, and hypertension seem to be more susceptible to DA. Since it penetrates the placental membrane, its foetal effects are detrimental [9]. Eight DA isomers (DA A-H) and one diastereomer (C5`, or isoDA) have been identified so far, all of them found to be less potent than DA itself [11].

The most common DA vectors are filter-feeding marine organisms such as shellfish, especially mussels, razor clams, and scallops, but DA was also found in crustaceans, octopods, cuttlefish, krill, anchovies, sardines, and planktivorous fish [9,10]. There are differences in the occurrence and concentration of DA between species, even when the latter are exposed to the same environmental influences [12,13,14,15]. These differences are attributed to the differences in DA absorption and depuration rates [16]. Some species such as *M. edulis* [17], *M. galoprovincialis* [18], *Mesodesma donacium* [19], and *Argopecten purpuratus* [20] depurate DA very quickly, unlike *Siliqua patula* [21] and *Pecten maximus* [22], which retain this phycotoxin for a long period of time. Even though DA seems to be non-toxic for shellfish, it exerts some adverse effects on bivalves, such as oxidative stress, mitochondrial dysfunction [23], DNA damage [24], haemocyte changes [25], shell closure, haemolymph acidosis, hypoxia [26], negative impact on larval [27] and adult growth rate, and the survival of some shellfish species [28].

This study aimed at determining the differences in occurrence and accumulation of DA in European oysters (*Ostrea edulis* Linnaeus, 1758), Queen scallops (*Aequipecten opercularis* Linnaeus, 1758), and edible ascidians of the *Microcosmus* spp. originating from the same harvesting area in the Northern Adriatic Sea, as well as at establishing seasonal differences in DA accumulation. Since concentrations far below the maximum permitted level (MPL) of 20 mg DA per kg were expected, the quantification was performed using ultra-performance liquid chromatography–tandem mass spectrometry (LC-MS/MS) preceded by derivatization with dansyl chloride (DNS-Cl) in order to achieve the lower limit of detection (LOD), as compared to the reference high-performance liquid chromatography with diode array detection (HPLC-DAD) method. The results may be of use in the identification of sentinel species suitable for the monitoring of DA level in seafood. Given that data on the occurrence of phycotoxins in European oysters, Queen scallops, and ascidians originating from the Northern Adriatic Sea are scarce, this study may contribute to shellfish-consumption-related consumer risk assessment and the promotion of ascidians as an alternative food source.

## 2. Results and Discussions

### 2.1. Validation of Analytical Method

The analysis of 20 blank samples did not reveal any interference at DA retention time. LOD and LOQ values determined using blanks approach described by Wenzl et al. [29] were similar for all investigated species and approximated to 0.3 and 1 µg/kg, respectively. During validation, the certified reference material CRM-FDMT1 was used for method recovery check-up. The obtained recovery ranged from 87% to 108%, with the average value of 98%. Repeatability, in-house reproducibility, recovery, and measurement uncertainty results are given in Table 1. Relative standard deviations of repeatability and in-house reproducibility were found to be below 15%, with the average recovery of 94%. These results are compliant with the validation requirements laid down under the Commission Decision 2002/657/EC, speaking to the suitability of this method for DA quantification in the investigated species.

As a part of the validation process, the matrix effect was evaluated due to the possible lack of response or an enhanced response (ion suppression or ion enhancement) in MS/MS detection, caused by coeluting matrix components that can affect the method performance [30,31]. When comparing the slopes of calibration curves of derivatized matrix-matched standards to the derivatized standards prepared in solvent, the difference in ratio of each of those curves higher than 10% as ion suppression was recorded, common for LC-MS/MS, especially if electro-spray ionisation (ESI) is engaged in the positive ionisation mode [32]. A significant difference was found between the slope of the calibration curve prepared with an ascidian blank matrix and the other two shellfish matrices, so, in order to minimize the matrix effect, matrix-matched calibration prepared with blank extracts of each and every investigated species was used for quantification.

### 2.2. Domoic Acid in Bivalves and Ascidians

The obtained results show the presence of DA in all investigated species. It was present in 36% of the samples in concentrations significantly lower than the MPL of 20 mg/kg, set out under the Regulation 853/2004/EC [33]. Bivalve and ascidian samples under this study represent a natural three-species population, which grew in the same area of the Northern Adriatic Sea, hence influenced by the same environment. In this way, inter-species differences caused by various environmental conditions seen in different locations were minimized. Table 2 presents the mean values of DA quantified in samples collected during the entire course of the study. The chromatogram of the sample harbouring the highest DA concentration is presented in Figure 1.

Statistical analysis revealed a difference in DA accumulation between the species (*p* < 0.05). Ascidians and European oysters showed no significant mutual difference in this regard (*p* = 0.210), whilst the difference between Queen scallops and the other two species—ascidians and oysters—was proven significant (*p* < 0.001). Queen scallops showed the greatest DA accumulation, with 57% of positives and a mean DA concentration 2.3-fold higher than that in European oysters. As compared to oysters, DA was detected in almost twice as many ascidian samples, but in lower mean and maximum concentrations.

The presence of DA in bivalve and ascidian species investigated in this study and the demonstration of differences between their DA accumulation are in line with the previous results obtained for naturally contaminated bivalves gathered from various world seas. Differences in DA accumulation between marine species collected at the same time at the same locality were observed by Ujević et al. [12] in rough cockle *Acanthocardia tuberculata* and smooth clam *Callista chione* collected from the Cetina River estuary in the Central Adriatic Sea. In rough cockle, DA was found from April through July 2009 in the maximum concentration of 770 µg/kg, while smooth clam was DA-positive in concentration of 280 µg/kg only on a single occasion in April of that year. Bouchouicha-Smida et al. [13] found a greater DA accumulation in oysters *Ostrea edulis* (420–1040 μg/kg) collected from the southwest Mediterranean in August and December 2009, as compared to simultaneously collected *M. galloprovincialis* (130–860 μg/kg). Takata et al. [14] determined the maximum DA level of 88,160 μg/kg in *Spondylus squamosus* gathered from the Philippines in March 2006, while other co-occurring species (*Perna viridis*, *Anadara antiquata*, *Chama iostoma*, *Chama lazarus*, *Atrina vexillum*, *Placuna sella*, *Hyotissa hyotis*) contained DA levels of around 1000 μg/kg. Picot et al. [15] found that the dominant DA vector is cockle *Cerastoderma edule*, in which DA was detected in the highest concentration of 3710 μg/kg. Carpet shells of the *Ruditapes* spp. harboured DA in the concentration of 2580 μg/kg, while razor clams (*Ensis* spp. or *Solen* spp.) contained 2520 μg/kg of DA. The latter two accumulated more DA than oysters *Crassostrea gigas* (1730 μg/kg) and mussels *M. edulis* (1040 μg/kg). All of the samples referred to above were collected simultaneously on the French Atlantic coast from June 2009 to June 2010.

The differences elaborated above are species-specific and can be attributed to different accumulation [16,34,35] and elimination rates [18,22,36], gut assimilation [16,37], tissue distribution [19,20,22], and environmental conditions [38]. DA depuration in bivalves is usually explained by two kinetic models [20]. The single-compartment model is characterized by an exponential DA decrease during the entire depuration period and describes DA depuration kinetics in *M. edulis* [17], *P. magellanicus* [39], and *P. maximus* [22]. The two-compartment model is characterized by rapid initial toxin elimination followed by a period of slower elimination. This pattern can be seen in *M. galloprovincialis* [18], *C. virginica* [36], and some other species. Due to the dissimilar depuration kinetics, differences are also seen between species belonging to the same family. For instance, in the *Argopecten purpuratus* scallop, DA acquired by the digestive gland is quickly transferred between the organs and rapidly released into the environment [20]. In another scallop, *Pecten maximus*, the majority of the toxin is accumulated in the digestive gland, the redistribution is limited, and the depuration is very slow [22,37].

According to Vale and Sampayo [40], carpet shell and common cockle contain higher DA concentrations than mussels and oysters. In their research, Mafra et al. [34] found that oysters have a lower capacity for DA accumulation as compared to mussels. This observation was explained by low oyster toxin intake due to the low clearance rate and the capability of selective rejection of *Pseudo-nitzschia* cells in pseudo-faeces. Faster DA elimination seen in some species such as *M. edulis* can also be explained by the presence of DA-utilizing bacteria capable of DA degradation in gut flora [41]. On the other hand, in species that retain DA for a longer period, such as the *Placopecten magellanicus* scallop, those bacteria are rarely found [41]. DA availability is also of great importance—since scallops mostly accumulate DA in their digestive gland, it becomes inaccessible to bacteria [41].

Furthermore, since the species under this study are all suspension feeders that utilise detritus as a food source [42,43,44], lower accumulation of DA in ascidians and oysters might be related to the lack of particle-sorting mechanism [45] and the dilution of food in the gut when feeding on concentrated high-particle detritus [46]. Newer research shows that some large microalgae cells can evade capture by ascidians due to some of their traits, such as the thick bio-mineralised armour of diatoms [47]. Similar findings are reported by Barillé and Cognie [44], who found a significant post-extracellular digestion survival of diatoms in pseudo-faeces and faeces of the Pacific oyster *Crassostrea gigas* regardless of cell size. The above is favoured by the dilution of food particles in the digestive system, arising on the grounds of addition of inorganic matter intended to create natural feeding conditions, and possibly also by the resistance of diatom cell walls to digestion [48]. Differences in DA absorption and excretion mechanisms seen in shellfish are investigated on the molecular level as well. Trainer and Bill [21] suggested the presence of high affinity–low capacity and low affinity–high capacity glutamate receptors that prevent DA intoxication of the Pacific razor clam *Siliqua patula*, which could be responsible for binding and retaining DA for a long period of time. Mauriz and Blanco [49] found DA in the King scallop (*Pecten maximus*), mostly free in cytosol in a soluble form. Nevertheless, this species retains DA for a long period of time, which was explained by the lack of membrane transporter necessary for DA to pass the plasma membrane. Even though mussels are commonly used for phycotoxins’ monitoring purposes, DA concentrations commonly present in them rarely make them the most intoxicated species [13]. Based on the data provided by this research, it can be concluded that, due to their longer DA retention and higher accumulation rate, Queen scallops are sentinel species more suitable for DA monitoring than oysters. Even though the presence of hydrophilic paralytic shellfish toxins and ASP phycotoxins in ascidians has been recorded, even in high concentrations that could pose a threat to human health [50,51,52], data provided by this study reveal that ascidians accumulate DA in significantly lower concentrations than oysters and scallops.

Findings of low-level presence of DA in shellfish and ascidians, yielded by this study, are in line with the research done in the Adriatic Sea and the Mediterranean area. DA in the maximum concentration of 2500 μg/kg was determined in Italian mussels in 2000 [53], as well as in different parts of the Adriatic Sea in concentrations far below the regulatory limit [54]. In 2005, DA was detected for the first time in Croatian mussels bred on the west side of the Istrian peninsula in concentrations of up to 872 μg/kg in *M. galloprovincialis* [55]. Within the 2006–2008 timeframe, DA was detected on several occasions in four species (Proteus scallop, Mediterranean scallop, European flat oyster, and blue mussel) bred and harvested in the Central and Northern Adriatic, in the highest concentration of 6549 μg/kg in *M. galloprovincialis* samples collected from the Central, along with 1657 μg/kg in *P. jacobeus* samples collected from the Northern Adriatic [56]. In 2007, Rijal Leblad et al. [57] reported the highest DA concentration of 4900 μg/kg found in sweet clam (*C. chione*) and 2110 μg/kg found in tuberculate cockle (*A. tuberculata*) originating from the Mediterranean coast of Morocco. The highest level of DA detected in Tunisian oysters was 1040 μg/kg [13]. Furthermore, the maximum concentrations of 860 μg/kg and 800 μg/kg were recorded in mussels from Tunisia and France, respectively, while the concentration found in mussels harvested in Spain and Portugal approximated to 5000 μg/kg [58].

Environmental conditions such as temperature, salinity, and food availability differ over the seasons and may influence the physiological status of shellfish, which in turn affects DA depuration kinetics [20]. Blanco et. al. [18] found that the decrease in salinity reduces DA depuration rate in mussels, while changes in water temperature and body weight have no significant effect. Positive correlation between environmental temperature and DA depuration rate was established in *P. maximus* [38]. In their research of gut passage time of microalgae in mussels and Pacific oysters, Guéguen et al. [48] found no significant effect of algal concentration on passage time in oysters, but did prove the impact of environmental temperature on cell lysis in the digestive tract, presumably linked to the higher activity of the digestive enzymes witnessed at higher temperatures.

This research revealed the maximum DA concentrations in Queen scallops and ascidians to be witnessed in the late autumn. In ascidians, significantly higher DA concentrations were found in December (*p* < 0.001), while in scallops, this was the case in both November and December (*p* = 0.049). In spring and summer, DA concentrations decreased. A significant month-to-month difference in DA concentrations (above LOD) was not seen in oysters (*p* = 0.868) but was evidenced in ascidians and scallops. Figure 2 shows the mean DA concentrations detected monthly in bivalves and ascidians during the one-year study, together with the pertaining standard errors.

These results conform to those of Stonik and Orlova [59], who reported the presence of DA in Pacific mussels (*M. trossulus*) from October 2009 to February 2010 in concentrations of up to 100 μg/kg. From January to March 2010, DA was found in *Mizuhopecten yessoensis* scallops collected in the Vostok Bay in the Sea of Japan, with the highest concentration of 90 μg/kg determined in January. Mussel samples collected during the year were DA-free in spring and summer. The findings of this research do not coincide with those of Ljubešić et al. [55] and Ujević et al. [56], who also reported on DA occurrence in the Northern Adriatic Sea. The authors quoted above detected DA in shellfish samples during the warmer part of the year. Ljubešić et al. [55] detected DA in *M. galloprovincialis* from April until October in concentrations ranging from 97 μg/kg to 872 μg/kg, whilst Ujević et al. [56] detected DA from July until November in *M. galloprovincialis*, *F. proteus* and *P. jacobaeus* in the highest concentration of 1657 μg/kg, measured in *P. jacobaeus* in July. Of note, the latter authors reported the maximum DA concentration of 6549 μg/kg in *M. galloprovincialis* from the Central Adriatic in February 2006, when the lowest sea temperatures in the eight-year period were recorded. One of the possible reasons for differential accumulation of DA in the investigated species could be the discordance in their spawning periods. High energy consumption during the spawning period has a negative impact on the condition index and the tolerance to environmental stress [60,61,62], especially in broadcast spawners such as scallops [63,64]. Since the reproduction pattern is influenced by species characteristics, environmental conditions, and the location [65,66], further research on reproductive cycles of bivalves and ascidians originating from the eastern coast of the Northern Adriatic Sea would be very valuable.

Even though all concentrations measured in this and the aforementioned studies were far below the MPL, it can be assumed that DA may occur in shellfish during any season, even in concentrations above the MPL. This was the case in France, where the maximum DA concentration determined in *D. trunculus* in spring 1999 equalled to 3200 μg/kg, while the next year, during the same period and in the same region, it increased to 53,000 μg/kg [67].

## 3. Conclusions

Domoic acid was detected in trace levels in three investigated species—European oysters (*O. edulis*), Queen scallops (*A. opercularis*), and ascidians of the *Microcosmus* spp., collected on the eastern coast of the Northern Adriatic Sea throughout the year. Queen scallops accumulated DA more often and in higher concentrations. The highest DA concentrations were found during the colder parts of the year, that is, in ascidians significantly higher in December and in scallops significantly higher in November and December. According to our data, there exists an interspecies variability in DA accumulation, occurrence, and seasonality. The obtained data could be of great value in planning phycotoxin monitoring programmes, in particular at locations at which multiple bivalve species are harvested. This study indicates the need for further research on DA presence in bivalves and other species originating from the eastern coast of the Northern Adriatic Sea that can act as DA vector to humans. Future research should include studies of different abiotic factors that can affect DA accumulation in these species.

## 4. Materials and Methods

### 4.1. The Collection of Samples

Samples were obtained from the same harvesting area in the waters of the northwest coast of the Istrian peninsula, Croatia (Φ 45°31′30″ N λ 13°27′18″ E). The Istrian peninsula is situated on the eastern coast of the Northern Adriatic Sea, i.e., in the mostly oligotrophic Adriatic region influenced by the eastern current, oligotrophic karstic rivers, and the northeaster wind—the bora [68]. Samples were collected from April 2018 until March 2019. In total, 46 European oyster, 53 Queen scallop, and 107 ascidian samples were collected during a one-year period. Shellfish were collected from the seabed using dredges; ascidians were also collected using hand tools. Upon collection, samples were immediately transported to the laboratory and extracted for subsequent analysis within 24 h upon arrival. If the analysis was not possible straightaway, the extracts were stored at −18 °C.

### 4.2. Chemicals and Analytical Standard

The certified CRM-DA-g calibration solution (331.9 ± 8.0 μmol/L), CRM-FDMT1 (containing 126 ± 10 mg/kg of DA + 5`-*epi*-DA based on the mass of the freeze-dried powder), and toxin-free control mussel tissue matrix CRM (CRM-Zero-Mus) were obtained from the National Research Council Canada’s Institute for Marine Bioscience (Halifax, NS, Canada). Chemicals and solvents used were of high and LC-MS purity grade, respectively.

### 4.3. Sample Preparation

Sample extraction and quantification were carried out using dansyl chloride derivatization and LC-MS/MS according to the method described by Beach et al. [69]. As for the ultra-trace analysis, an alternative procedure was used. After DA extraction from bivalve and ascidian soft tissue (100 g) using 50% aqueous methanol solution, extracts underwent a clean-up on a strong anion-exchange SPE cartridge. The eluate was concentrated using a nitrogen stream and reconstituted in a 50% aqueous acetonitrile solution. The reconstituted samples were derivatized with DNS-Cl using equal proportions of the reconstituted extract, 150 mM borate buffer aqueous solution, and 5.5 mM DNS-Cl acetonitrile solution. DNS-Cl surplus was removed by liquid–liquid extraction with hexane and analysed using LC-MS/MS. The same procedure was applied to the calibration standards.

### 4.4. LC-MS/MS Analyses

The quantification of DA was performed using a 1290 Infinity UPLC (Agilent Technologies, Singapore) coupled with G6460 Electrospray Ionisation Triple Quad Mass Spectrometer (Agilent Technologies, Waldbronn, Germany). Chromatographic separation was done on a Zorbax SB-C18 RRHT 2.1 × 50 mm, 1.8 μm column with SB-C18 2.1 × 5 mm, 1.8 μm precolumn (Agilent Technologies, Santa Clara, CA, USA), thermostatted at 40 °C. The mobile phase A consisted of 0.2% (*v*/*v*) formic acid in water, while the mobile phase B was composed of 0.2% (*v*/*v*) formic acid in acetonitrile. An isocratic elution with a flow of 0.5 mL/min was applied, with the 30.0%-involvement of the organic mobile phase B, and the injection volume of 5 μL. For the sake of quality control, negative (blank) and positive (spiked) samples were analysed with every sample batch. Quantification was done using an external matrix-matched calibration curve ranging from 1 to 1500 µg/kg. The concentrations were corrected using the average recovery values obtained by validation.

### 4.5. Validation

The method was validated in line with the Commission Decision 2002/657/EC [70], applying the InterVAL Plus Software Version 3.4.0.4 (quo data, Gesellschaft für Qualitätsmanagement und Statistik GmbH, Dresden, Germany) for experimental design and calculation of validation parameters. Repeatability and in-house reproducibility were determined by analysing DA in 128 spiked samples consisting of an equal proportion of three bivalve species and ascidians (even though Mediterranean mussels were not under study, they were included in the validation). The limit of detection (LOD) and the limit of quantification (LOQ) were determined in each matrix using the blanks approach, the LOQ thereby being calculated as 3.3 times LOD [29]. To determine the presence of the matrix effect, derivatized standard solutions in water/acetonitrile (1:1) were injected in parallel with derivatized extracts of shellfish and ascidians spiked with the standard DA solution. The value of the calibration curve slope in solvent was compared to the value of slopes of matrix-matched calibration curves. The same comparison was also performed between different matrices.

### 4.6. Statistical Analysis

Statistical analysis was performed using the SPSS Statistics Software 22.0 (IBM, Armonk, NY, USA). The results were tested for the normality of their distribution using the Shapiro–Wilks test. In order to determine the statistical significance (*p* < 0.05) of the differences in DA concentrations in different species and sampling months over a one-year period, the non-parametric Kruskal–Wallis test was used.

## Figures and Tables

**Figure 1 toxins-14-00033-f001:**
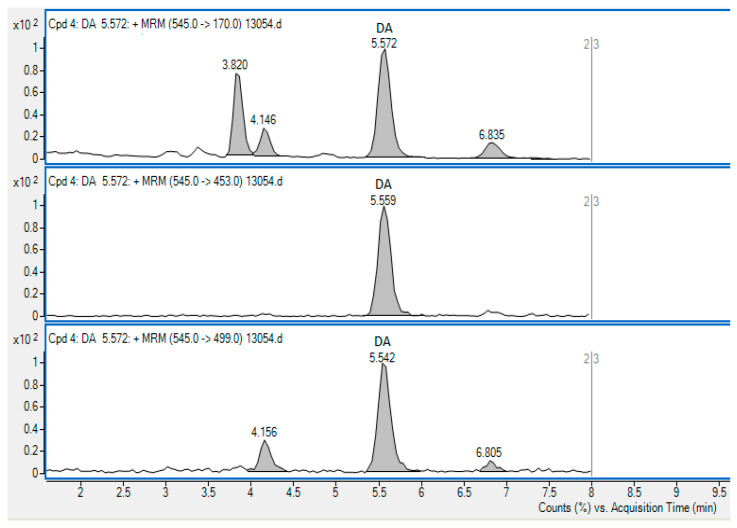
The chromatogram showing the method of SRM transitions used for the detection of DA in the Queen scallop sample containing DA in the concentration of 809.5 µg/kg. The 545→170 transition (precursor *m*/*z*→product *m*/*z*) was used for quantification.

**Figure 2 toxins-14-00033-f002:**
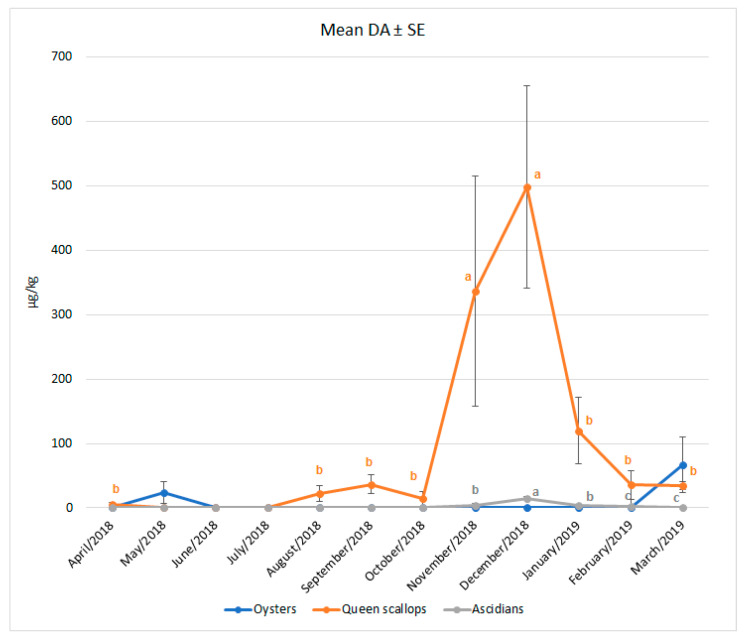
Monthly occurrence of DA in European oysters, Queen scallops and ascidians, detected from April 2018 until March 2019. ^a,b,c^ values within the same line (species) tagged with a different superscript differ significantly (*p* < 0.05). Vertical bars denote the standard errors.

**Table 1 toxins-14-00033-t001:** Repeatability, in-house reproducibility, recovery, and measurement uncertainty established for DA determined in Mediterranean mussels, European oysters, Queen scallops, and ascidians ^a^.

Calibration Level (μg/kg)	s_r_ (μg/kg)	Rs_r_ (%)	s_wR_ (μg/kg)	Rs_wR_ (%)	Recovery (%)	Δ (%)
50	4.3	9.5	5.3	10.6	93	18
250	19.9	8.0	21.2	8.5	94	14
500	39.5	7.9	41.6	8.3	94	14
1000	78.9	7.9	7.9	8.3	94	14

DA (domoic acid), s_r_ (repeatability standard deviation), R_Sr_ (relative repeatability standard deviation), s_wR_ (in-house reproducibility standard deviation), R_SwR_ (relative in-house reproducibility standard deviation), Δ (extended relative measurement uncertainty, k = 1.645); ^a^ The total number of samples (*n* = 128) consisting of an equal share of the four species (a quarter of each).

**Table 2 toxins-14-00033-t002:** Domoic acid in bivalves and ascidians originating from the same harvesting area in the Northern Adriatic Sea.

Species	n	% of Positives *	Mean ± SE of Positives * (μg/kg)	Maximum (μg/kg)
European oysters	46	17	65.6 ± 10.3 ^b^	212
Queen scallops	53	57	153 ± 32.1 ^a^	810
Ascidians	107	33	5.5 ± 0.6 ^b^	24.3

* values above LOD; ^a,b^ values within a column tagged with a different superscript differ significantly (*p* < 0.05). SE (standard error).

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
