# Peer review of "Occurrence and Seasonal Monitoring of Domoic Acid in Three Shellfish Species from the Northern Adriatic Sea"

_toxins, 2022, doi:10.3390/toxins14010033_

Round 1

Reviewer 1 Report

The growing efforts to detect and monitoring new amnesic shellfish poisoning (ASP) outbreaks worldwide are strictly necessary to avoid the severe negative impacts of DA-producing species from the genus Pseudo-nitzschia. This work provides information about some shellfish species that could be potential vectors of ASP to humans in the northern Adriatic Sea. This knowledge is essential to strengthening ASP-monitoring networks around the world. Nonetheless, the feasibility of the results in this study is not conclusive. The reading of this manuscript is complicated and confusing, and serious misconceptions are found throughout the whole manuscript. Furthermore, a significant improvement in the use of the English language is required and major corrections are necessary before it is considered suitable for publication.

  • Line 2-3: the title of the manuscript is not convincing, it is not consistent or homogeneous, and does not reflect the essence of the results. Only the genus of the ascidians is specified, it does not appear to be an important accumulation of toxin in any of the three monitored shellfish species, and it does not mention where the study was done. I suggest changing it to "Occurrence and Seasonal Monitoring of the amnesic toxin Domoic Acid in three shellfish species from the Northern Adriatic Sea"

Abstract

  • Line 4: ascidians are also shellfish species, I suggest specifying "bivalves" and "ascidians".
  • Line 6: replace the article "a" with "the" before potent. Please consider adding the abbreviations "DA" for domoic acid, and "ASP" for amnesic shellfish poisoning to avoid typing the full names every time it is required through the document. Also, specify that ASP occurs in humans.
  • Line 8: replace "hydrophilic marine biotoxin" with "phycotoxin".
  • Line 10: add the article “the” before “Northern Adriatic” and “Sea” after. The determination of what? Of the toxin? the most appropriate term is "quantification". Please be as specific as possible across the manuscript.
  • Line 12: these methodological details are unnecessary in this section of the manuscript, when mention LC-MS/MS as a DA-detection technique it is understood that the amounts of the toxin were low.
  • Line 13: replace the article "the" with "a" before "maximum".
  • Lines 14-15: in which species were these maximum toxin values recorded? Moreover, move the article "the" before "occurrence". There were only differences between the investigated species? there are no seasonal differences? Please be specific.
  • Line 15: "acid" does not seem like an appropriate manner to refer to this toxin, please replace it with "DA".
  • Line 16: the species do not have a "preference" for DA, they accumulate the toxin in a differentiated way. Please rewrite this sentence and remove "as" before "compared".
  • Line 19: keywords do not reflect the content and contributions of this paper. I suggest removing "LC-MS / MS" and "shellfish" and adding, "Amnesic Shellfish Poisoning", "seasonal monitoring", and "bivalves".
  • Lines 20-21: What species are proposed as sentinels for AD monitoring?
  • Line 22: I did not see an economic and bromatological analysis of the ascidians in the results section to strengthen this statement. I suggest removing the words "economic”, “nutritional”, and just saying that ascidians are proposed as a safe alternative for human consumption, and specify what evidence supports this suggestion.

Hereafter corrections in grammar, syntax, semantics, vocabulary, and spelling will not be provided. This document requires a deep and important improvement of the English language, there is a profound misuse of specialized scientific vocabulary. All the mentioned above makes the reading of this manuscript very complicated.

Introduction

This section is very confusing, as its name implies, it should introduce the reader to the problem, the background on the phenomenon, and the objectives of the investigation, but it is far from that. The ideas are mixed, there is no logical sequence/order in the structure of this section. References are missing for many of the statements made throughout this section. Several unnecessary details are mentioned, and supporting knowledge that is needed is not well presented or is completely absent.

  • Lines 25-32: This paragraph is totally out of context. This is not a work on food safety/security, human nutrition, or sustainability. There is no evidence throughout this manuscript to support the assumption that ascidians are a food source suitable for human consumption.
  • Lines 33-38: Why only details about ascidians are provided? Reading the first part of the introduction gives the impression that the paper is focused only on the genus Microcosmus sp. and the rest of the shellfish are mentioned just in the final paragraph of this section. This is very confusing.
  • Lines 39-55: These paragraphs are confusing, they need to be rewritten. The ideas about physical-chemical properties of the toxin, DA toxicology, clinical-symptomatology of intoxication, and the main species responsible for DA-production are mixed and the message that the author wants to deliver to the reader is not understood.
  • Lines 44-46: This ASP outbreak in Canada and its tragic consequences are mentioned in thousands of publications on the subject of DA, there is no need to keep repeating it. Instead, it is necessary to provide a background on toxigenic and non-toxigenic blooms of diatoms of the genus Pseudo-nitzschia, or details on previous knowledge of accumulation of this toxin in other species of marine invertebrates in this study area.
  • Lines 56-58: this is probably the most important section of the introduction of this paper, since this is a work focused on the occurrence and monitoring of DA in shellfish, and only three lines are provided to describe it. Please include more details on which species accumulate more DA than others, emphasizing in DA toxicokinetics and from an interspecific point of view. Furthermore, provide scientific names and specific references for each of these species.
  • Lines 59-71: It is not necessary to give details about the DA-quantification method in this section; these can be detailed in the corresponding materials and methods part. Although it is not very appropriate to give conclusions at the end of the introduction, it seems that these statements are far from what the evidence of the results can support. There are no data presented in the background knowledge about the amounts of DA that these shellfish species are capable of accumulating in this area, the fact that they are not heavily contaminated does not mean that they are not potential vectors of the toxin towards humans. I suggest deepening the bibliographic review on this.

Results and discussion

As far as I know, the guide for authors of this journal requests sections of results and discussion should be separated for original articles. Apart from the mentioned above, the results have to be simplified and the statistical analyzes have to be strengthened. The bibliographic review to contrast the results of this work is very poor and needs to be improved.

  • Lines 73-101: this is not a work on a novel DA-quantification method, details on repeatability, in-house reproducibility, recovery, measurement uncertainty, and matrix effect are unnecessary unless requested by a reviewer, or they could be easily presented as supplementary material. Moreover, Table 1 is not described at all, no comparisons of the LOD and LOQ between the analyzed species are provided.
  • Lines 102-112: this is very confusing and difficult to understand. Why talk about the samples individually? Why show percentages of positives, medians, etc.? Please simplify table 3, there is no need to include the number of samples by species that was already specified in the text. There is no need to present positives; the percentage of samples where the DA-burdens were above the LOD can be easily described in a few words within the text. There is no need to present the medians if the means are already provided. In any case, present these data in a box-plot graph.

The fact of presenting the mean and maximum toxin burdens in µg kg-1 creates enormous confusion since in the abstract it was stated that the maximum amount of toxin detected in the samples was 0.8 mg DA kg-1. Please be consistent and homogeneous and present these results of toxin quantification as mg kg-1 or µg g-1.

The inter-individual variation is enormous, please calculate the standard error and present it instead of the standard deviation as a dispersion measurement. If the results are shown in a table, then include a column for statistical analysis, where the result of the statistic-test, degrees of freedom, residuals, and probability values ​​are indicated.

  • Lines 117-120: there are no significant differences in the means of the DA-amounts between oysters and ascidians? That is not possible. Make the description simpler, saying that significantly highest amounts of DA were quantified in opercularis compared to the other two species, rather than making paired descriptions that are confusing.
  • Lines 120-122: For the last time, some species have no more affinity to DA than others. In this case, opercularis accumulated higher (P <0.05) amounts of DA than the other two species.
  • Line 122: Figure 1 showing the chromatograms is completely unnecessary and does not support this statement. It has to be removed.
  • Lines 122-124: on what result is this statement based? if it is in the percentage of samples above the LOD then this statement is wrong. Furthermore, this idea contradicts everything that has been mentioned above about proposing ascidians as a safe food source for humans.
  • Lines 129-132: this statement is wrong. I recommend a deepest review of the literature to realize that the intra and interspecific differences in DA-accumulation are very different in this work concerning to other studies in several coastal areas around the world.
  • Lines 129-146: If all these antecedents exist, why none is mentioned in the introduction of the work?
  • Lines 143-146: be more specific; provide details on DA-amounts for each species.
  • Lines 147-149: the reference that supports this statement is very ambiguous. Give specific references for each aspect. Additionally, to date, there is no evidence of DA-biotransformation in contaminated marine invertebrates.
  • Lines 149-151: provide specific references for each species.
  • Lines 151-153: this reference is not enough to support this statement. I suggest reviewing the work of Alvarez et al. (2020) to strengthen this idea.
  • Lines 153-156: mention the amount of DA between species; otherwise, it is very difficult to make these kinds of inferences and comparisons.
  • Line 172: specify how low concentrations of DA have been recorded in other shellfish species in the Adriatic Sea.
  • Lines 177-178: to which species do these amounts of DA correspond?
  • Lines185-187: an assumption like this cannot be made; it is too speculative since the phytoplankton community was not sampled at the same time as the shellfish. In any case, it is presumable.
  • Lines 199-203: DA occurs in which amounts in all these species?
  • Lines 205-217: for each reference, for each site, specify the amounts of DA measured in which species; otherwise, it is very difficult to contrast the results of this work.
  • Line 218: Table 4 is very difficult to read and understand. Please simplify and modify it with a detailed statistical analysis as suggested in Table 3.

Conclusions

  • Lines 223-231: This section reads more like a summary of results than conclusions. Please rewrite it emphasizing the biological interpretation of the results and the main contributions of the knowledge derived from this work for future research.
  • Lines 234-235: What other species should be considered as DA-sentinels in monitoring programs? What is the basis for saying this? this statement demerits all the work presented in this paper.

Materials and methods

  • Lines 238-248: specify the coordinates of the sampling site instead of providing oceanographic details about the area. The total shellfish sample quantity in kg does not provide any additional information. Instead, specify the average length (cm) and weight (g) of each species recorded during the annual sampling cycle.
  • Lines 250-295: This section is overwhelmingly long, it needs to be summarized and simplified. The protocol for sample preparation and toxin extraction is very confusing and different from the methods traditionally applied in these kinds of studies. This makes me deeply doubt the feasibility of the results.
  • Lines 296-331: it is unnecessary to give so much detail about the MS parameters and the validation of these analyzes. Please summarize this in a few lines.
  • Lines 333-334: these lines are incorrectly placed in this section, they should be in the DA-quantification section, and the origin of the DA standard solutions used to construct the calibration curve should be specified.
  • Lines 334-338: More details about the statistical analyzes used in this work are needed, such as what a priori tests were applied to measure the homogeneity of variances and the frequency distribution of the data? What post hoc test was used for the multiple comparisons of means/medians?

References

An extensive revaluation and replacement, as well as the deepest revision of other bibliographic references are necessary to strengthen this work.

Author Response

Comment:  The growing efforts to detect and monitoring new amnesic shellfish poisoning (ASP) outbreaks worldwide are strictly necessary to avoid the severe negative impacts of DA-producing species from the genus Pseudo-nitzschia. This work provides information about some shellfish species that could be potential vectors of ASP to humans in the northern Adriatic Sea. This knowledge is essential to strengthening ASP-monitoring networks around the world. Nonetheless, the feasibility of the results in this study is not conclusive. The reading of this manuscript is complicated and confusing, and serious misconceptions are found throughout the whole manuscript. Furthermore, a significant improvement in the use of the English language is required and major corrections are necessary before it is considered suitable for publication.

Answer: We are grateful to our esteemed Referee for the recognition and appreciation of the significance of this study. We made every effort to rewrite the original manuscript according to the provided comments and remarks and do hope that the quality of the manuscript has been significantly improved. The manuscript was thoroughly checked by a highly qualified linguist (a University Professor) having a vast experience in scientific writing.

Comment: Line 2-3: the title of the manuscript is not convincing, it is not consistent or homogeneous, and does not reflect the essence of the results. Only the genus of the ascidians is specified, it does not appear to be an important accumulation of toxin in any of the three monitored shellfish species, and it does not mention where the study was done. I suggest changing it to "Occurrence and Seasonal Monitoring of the amnesic toxin Domoic Acid in three shellfish species from the Northern Adriatic Sea"

Answer: The title was changed as suggested by the Referee, so as to reflect the study design and the essence of the results more accurately. The investigated aquatic organisms are specified as suggested, together with the study area.

Comment: Line 4: ascidians are also shellfish species, I suggest specifying "bivalves" and "ascidians".

Answer: We appreciate the Reviewer’s remark regarding the term “shellfish”. The suggested replacements have been made throughout the manuscript.  

Comment: Line 6: replace the article "a" with "the" before potent. Please consider adding the abbreviations "DA" for domoic acid, and "ASP" for amnesic shellfish poisoning to avoid typing the full names every time it is required through the document. Also, specify that ASP occurs in humans.

Answer: The article “a” was replaced with “the”. Abbreviations “DA” and “ASP” have been used throughout the manuscript, as requested. It is now clearly specified that ASP occurs in humans.

Comment: Line 8: replace "hydrophilic marine biotoxin" with "phycotoxin".

Answer: The term "hydrophilic marine biotoxin" was replaced with "phycotoxin" (Line 10).

Comment: Line 10: add the article “the” before “Northern Adriatic” and “Sea” after. The determination of what? Of the toxin? the most appropriate term is "quantification". Please be as specific as possible across the manuscript.

Answer: In Line 13, “the” was added before “Northern Adriatic” and “Sea” after. In Lines 13, 89, 112, 409 and 437, the word “determination“ was replaced with “quantification”.

Comment: Line 12: these methodological details are unnecessary in this section of the manuscript, when mention LC-MS/MS as a DA-detection technique it is understood that the amounts of the toxin were low.

Answer: In line with the Referee’s opinion, details descriptive of the method have been omitted (Line 15), except for the information that the procedure included DNSL-Cl derivatisation. This is perceived by the authors as the fact informative of the method appropriate for the quantification of trace amounts of various toxins.

Comment: Line 13: replace the article "the" with "a" before "maximum".

Answer: The article “the” was replaced with “a” before “maximum” (Line 16).

Comment: Lines 14-15: in which species were these maximum toxin values recorded? Moreover, move the article "the" before "occurrence". There were only differences between the investigated species? there are no seasonal differences? Please be specific.

Answer: The maximum toxin value was recorded in Queen scallops, as added in the revised body text (Line 17). The article “the” was moved as requested. Differences between species are now described in more details in Lines 17-21.

Comment: Line 15: "acid" does not seem like an appropriate manner to refer to this toxin, please replace it with "DA".

Answer: In Line 18, the word “acid” is now replaced with the full name “domoic acid”.

Comment: Line 16: the species do not have a "preference" for DA, they accumulate the toxin in a differentiated way. Please rewrite this sentence and remove "as" before "compared".

Answer: We appreciate the Reviewer’s remark on the inappropriate use of terminology. This sentence has been rewritten and the word "preference" is omitted, as is the word "as" before "compared" (Lines 20-21).

Comment: Line 19: keywords do not reflect the content and contributions of this paper. I suggest removing "LC-MS/MS" and "shellfish" and adding, "Amnesic Shellfish Poisoning", "seasonal monitoring", and "bivalves".

Answer: We appreciate the Reviewer’s constructive feedback aimed at improving this article. The keywords "LC-MS/MS" and "shellfish" have been deleted and replaced with "seasonal monitoring" and "bivalves". However, "Amnesic Shellfish Poisoning" was not added, because the authors are of the opinion that the keyword “domoic acid” better represents the content of this manuscript. Namely, our study was focused on phycotoxin monitoring and quantification rather than DA toxicity.

Comment: Lines 20-21: What species are proposed as sentinels for AD monitoring?

Answer: To the best of our knowledge, no shellfish species have been officially proposed as sentinels for DA monitoring, either in literature or within any legislative frame. Nevertheless, the species most often used for phycotoxin monitoring are mussels, which is the case even in areas in which various shellfish species are farmed. Given that various species accumulate DA in a differentiated way, while monitoring programmes tend to use only a single species, as detailed above, it was our intent to point out that further research targeted at more effective monitoring designs embracing more than one sentinel should be done.

Comment: Line 22: I did not see an economic and bromatological analysis of the ascidians in the results section to strengthen this statement. I suggest removing the words "economic”, “nutritional”, and just saying that ascidians are proposed as a safe alternative for human consumption, and specify what evidence supports this suggestion.

Answer: We accepted the Reviewer’s suggestion and removed the words "economic” and “nutritional”. In many parts of the world, ascidians are an integral part of human diet and are considered a delicacy, since they provide a number of bioactive components (Vafidis et al. 2008; Meenakshi et al. 2012; Lambert et al. 2016). Even though economic and bromatological analyses were not carried out, it was our wish to briefly present the nutritive value of these organisms, which is both under-recognised and under-appreciated, so that bivalve consumption currently prevails. Given that risk assessment relative of this species is not a part of this study design, we are of the opinion that we are not yet in position to claim that ascidians represent a safe alternative for human consumption (Line 26).

Comment: Hereafter corrections in grammar, syntax, semantics, vocabulary, and spelling will not be provided. This document requires a deep and important improvement of the English language, there is a profound misuse of specialized scientific vocabulary. All the mentioned above makes the reading of this manuscript very complicated.

Answer: As stated above, the manuscript was thoroughly checked by a proficient linguist experienced in scientific writing, hopefully to the Referee’s satisfaction. Spelling differences might have arisen from the fact that the manuscript is written in UK, not US English. The use of technical terms is the sole responsibility of the authors, who made every effort to comply with the Reviewer’s suggestions.

Comment: Lines 25-32: This paragraph is totally out of context. This is not a work on food safety/security, human nutrition, or sustainability. There is no evidence throughout this manuscript to support the assumption that ascidians are a food source suitable for human consumption.

Answer: Due to its multiple health benefits, seafood is a very desirable component of consumer diet. That was the exact reason for mentioning seafood in the introductory part, so as to explain why our study targets these organisms in particular (Lines 30-35).

Comment: Lines 33-38: Why only details about ascidians are provided? Reading the first part of the introduction gives the impression that the paper is focused only on the genus Microcosmus sp. and the rest of the shellfish are mentioned just in the final paragraph of this section. This is very confusing.

Answer: As explained already, since the ascidians are under-appreciated as a species capable of favourably contributing to human diet, it was our intent to elaborate their value in a few sentences. Nevertheless, to avoid confusion and the wrong impression that the paper is focused on this genus only, the pertaining text in Lines 38-39 was deleted. The rest of the paragraph describes filter feeders - bivalves and ascidians.

Comment: Lines 39-55: These paragraphs are confusing, they need to be rewritten. The ideas about physical-chemical properties of the toxin, DA toxicology, clinical-symptomatology of intoxication, and the main species responsible for DA-production are mixed and the message that the author wants to deliver to the reader is not understood.

Answer: Paragraphs concerning physical-chemical properties of the toxin, DA toxicology, clinical symptomatology of intoxication, and the main species responsible for DA-production are now rewritten in a manner easier to read and comprehend. After a few details on chemical properties and producing organisms (Lines 44-47), toxicity, clinical symptoms and the adverse impact on sensitive populations are described in short (Lines 47-57).

Comment: Lines 44-46: This ASP outbreak in Canada and its tragic consequences are mentioned in thousands of publications on the subject of DA, there is no need to keep repeating it. Instead, it is necessary to provide a background on toxigenic and non-toxigenic blooms of diatoms of the genus Pseudo-nitzschia, or details on previous knowledge of accumulation of this toxin in other species of marine invertebrates in this study area

Answer: We agree with the Referee, so that the sentences in Lines 58-61 concerning animal and human DA intoxication in Canada have been deleted. After mentioning the species involved in DA transfer to higher food chain instances, differences in accumulation of this phycotoxin between various species are discussed, followed by the description of adverse DA effects on bivalves (Lines 70-83).

Comment: Lines 56-58: this is probably the most important section of the introduction of this paper, since this is a work focused on the occurrence and monitoring of DA in shellfish, and only three lines are provided to describe it. Please include more details on which species accumulate more DA than others, emphasizing in DA toxicokinetics and from an interspecific point of view. Furthermore, provide scientific names and specific references for each of these species.

Answer: As stated in the answer to the previous comment, we have now described inter-species differences in DA accumulation in more detail (Lines 73-78) and have added scientific names of the species and supportive references. In order to make this section concise, differences in toxicokinetics are discussed under the Results and Discussion section.

Comment: Lines 59-71: It is not necessary to give details about the DA-quantification method in this section; these can be detailed in the corresponding materials and methods part. Although it is not very appropriate to give conclusions at the end of the introduction, it seems that these statements are far from what the evidence of the results can support. There are no data presented in the background knowledge about the amounts of DA that these shellfish species are capable of accumulating in this area, the fact that they are not heavily contaminated does not mean that they are not potential vectors of the toxin towards humans. I suggest deepening the bibliographic review on this.

Answer: Details of the method used are indeed given in the Materials and Methods section. Information on the method provided in the Introduction section is given to make it clear that this study was not a part of the routine phycotoxin monitoring, but an independent study targeted at the detection of DA trace amounts. Since the published data on the occurrence of DA in the investigated species originating from the same study location are scarce, especially when it comes to A. opercularis and Microcosmus spp., we were of the opinion that this research might be a valuable contribution. Even though data on phycotoxin monitoring in this geographical region are lacking, the research we have done insofar, and its preliminary results, suggested the possibility of DA presence in this region. These preliminary results also led us to believe that the levels of contamination are going to be low, which explains our selection of the quantification method.

Comment: As far as I know, the guide for authors of this journal requests sections of results and discussion should be separated for original articles. Apart from the mentioned above, the results have to be simplified and the statistical analyzes have to be strengthened. The bibliographic review to contrast the results of this work is very poor and needs to be improved.

Answer: We are most obliged to the Referee for all his/her comments, which we made an enormous effort to comply with. We also tried to shape the paper according to the instructions of the esteemed journal to the best of our abilities.

Comment: Lines 73-101: this is not a work on a novel DA-quantification method, details on repeatability, in-house reproducibility, recovery, measurement uncertainty, and matrix effect are unnecessary unless requested by a reviewer, or they could be easily presented as supplementary material. Moreover, Table 1 is not described at all, no comparisons of the LOD and LOQ between the analyzed species are provided.

Answer: We are of the opinion that validation of the method used in this study is an important part of our research since it aimed at demonstrating the suitability of the method for quantification of DA trace amounts. In line with the Referee’s comment and given that the LOD/LOQ values obtained for different species were similar, Table 1 has been removed, but the paper still contains brief information on the method and validation results (Lines 101-107).

Comment: Lines 102-112: this is very confusing and difficult to understand. Why talk about the samples individually? Why show percentages of positives, medians, etc.? Please simplify table 3, there is no need to include the number of samples by species that was already specified in the text. There is no need to present positives; the percentage of samples where the DA-burdens were above the LOD can be easily described in a few words within the text. There is no need to present the medians if the means are already provided. In any case, present these data in a box-plot graph.

Answer: Lines 134-145 have been changed in the manner the Referee will hopefully find easier to comprehend. The Median column has been deleted, which simplifies the Table, but we decided to leave the data on the number of samples, so as to make the Table clear on its own. Standard deviation has been replaced with standard error. Within the frame of this study, we chose to use the common manner of study results’ presentation, most often resorted to by other study authors investigating into the presence of other food contaminants, as well.

Comment: The fact of presenting the mean and maximum toxin burdens in µg kg-1 creates enormous confusion since in the abstract it was stated that the maximum amount of toxin detected in the samples was 0.8 mg DA kg-1. Please be consistent and homogeneous and present these results of toxin quantification as mg kg-1 or µg g-1.

Answer: For the sake of consistency and in line with literature sources, concentrations previously expressed as mg/kg are now expressed as µg/g.

Comment: The inter-individual variation is enormous, please calculate the standard error and present it instead of the standard deviation as a dispersion measurement. If the results are shown in a table, then include a column for statistical analysis, where the result of the statistic-test, degrees of freedom, residuals, and probability values are indicated

Answer: In line with the Referee’s opinion, standard error has been calculated instead of standard deviation. The values are given in Table 2. We agree with our esteemed Referee that the presentation of the results of statistical analysis, in specific that of the p-values, is of importance, but we fear that the display of each and every intercomparison in a single table would be quite confusing and redundant. Namely, the results in question read as follows: ascidians vs oysters, p=0.210; oysters vs queen scallops, p< 0.001; ascidians vs queen scallops, p< 0.001, and are given in the body text (Lines 150-154).  The degree of freedom is 1 (unexceptionally). Statistical significance provided in the Table is letter-tagged. Should the Referee be of the opinion that the display of p-values in the Table would not be complicated and puzzling and has a suggestion how to include it, we are more than willing to comply.

Comment: Lines 117-120: there are no significant differences in the means of the DA-amounts between oysters and ascidians? That is not possible. Make the description simpler, saying that significantly highest amounts of DA were quantified in opercularis compared to the other two species, rather than making paired descriptions that are confusing.

Answer: The statistics has been rechecked. Given the data distribution pattern, a non-parametric test comparing not means, but medians, was used, showing no statistically significant difference between oysters and ascidians (the medians are exactly the same). In line with the Referee’s request, the sentence was rewritten to be clearer (Lines 150-154) and provides p-values, i.e., the results obtained by the test referred to above.

Comment: Lines 120-122: For the last time, some species have no more affinity to DA than others. In this case, opercularis accumulated higher (P <0.05) amounts of DA than the other two species.

Answer: Once again, we appreciate the Reviewer’s remark on the inappropriate use of terminology. In Line 154, the word "affinity" has been removed from the text.

Comment: Line 122: Figure 1 showing the chromatograms is completely unnecessary and does not support this statement. It has to be removed.

Answer: We would appreciate keeping Figure 1 in the paper, since it presents the method SRM transitions used for the detection of DA. Instead, Table 5 under the Materials and Methods section could be removed in order to make the paper more concise and easier to read.

Comment: Lines 122-124: on what result is this statement based? if it is in the percentage of samples above the LOD then this statement is wrong. Furthermore, this idea contradicts everything that has been mentioned above about proposing ascidians as a safe food source for humans.

Answer: The Lines are rewritten to make it clear that DA was found in more ascidian than oyster samples, which were also DA- positive, but harboured DA in lower concentrations. (Lines 156-158). The statement that the ascidians are safe for human consumption has been omitted since no risk assessment was made or foreseen by the study design).

Comment: Lines 129-132: this statement is wrong. I recommend a deepest review of the literature to realize that the intra and interspecific differences in DA-accumulation are very different in this work concerning to other studies in several coastal areas around the world.

Answer: The statement was intended to emphasise that in this research, as well as in the research cited afterwards in Lines 169-172, DA was detected in samples collected at the same site, which were proven to differ as regards DA accumulation. The latter goes for the species farmed (grown) in the same environments. All of the above does not relate to the detected DA concentrations, but to the DA occurrence.

Comment: Lines 129-146: If all these antecedents exist, why none is mentioned in the introduction of the work?

Answer: In the revised manuscript, the studies quoted in Lines 172-191 are briefly mentioned in the Introduction section, as well (Lines 73-78).

Comment: Lines 143-146: be more specific; provide details on DA-amounts for each species.

Answer: Details concerning the study cited in Lines 184-190 have been added. Scientific names of simultaneously collected investigated species and peak DA concentrations quantified by LC-MS/MS have been added.

Comment: Lines 147-149: the reference that supports this statement is very ambiguous. Give specific references for each aspect. Additionally, to date, there is no evidence of DA-biotransformation in contaminated marine invertebrates.

Answer: The reference addressed by the Reviewer has been replaced with 10 new references better supporting each study aspect (Lines 192-194). We appreciate the Reviewer’s remark on the lack of evidence on biotransformation in contaminated marine invertebrates. Being aware of that fact, we have entirely deleted the corresponding text (Line 193,194).

Comment: Lines 149-151: provide specific references for each species.

Answer: Lines 149-151 are now rewritten to mention the kinetic models of DA depuration and species hosting the depuration that takes place in the manner described by the models. The pertaining references have been added, too (Lines 196-202).

Comment: Lines 151-153: this reference is not enough to support this statement. I suggest reviewing the work of Alvarez et al. (2020) to strengthen this idea.

Answer: We are most obliged to our Reviewer for the suggestion to review the work of Alvarez et al. (2020). This turned out to be a very valuable literature source that led us to other scientific reports written by the same and by other authors engaged in this line of research, which helped us strengthen our work. The pertaining part of the paper has been rewritten to explain the subject-matter in more detail (Lines 202-210).

Comment: Lines 153-156: mention the amount of DA between species; otherwise, it is very difficult to make these kinds of inferences and comparisons.

Answer: The work of Bouchouicha-Smida et al. (2015) can now be found in Lines 178-180, together with the maximal DA concentrations detected in the investigated species. As for the work of Vale and Sampayo (2012), we have commented on simultaneously collected and analysed shellfish samples of various sorts, originating from the same area. The obtained concentrations have been reported in form of a diagram, and a mutual comparison was made (Lines 213-214).

Comment: Line 172: specify how low concentrations of DA have been recorded in other shellfish species in the Adriatic Sea.

Answer: The wording “also in low concentrations” has been replaced with “in concentrations far below the regulatory limit” (Lines 260-261). We hereby refer to the review article authored by Arapov (2013), in which individual shellfish species and DA concentrations found in them were not reported.   

Comment: Lines 177-178: to which species do these amounts of DA correspond?

Answer: In Lines 266-267, species corresponding to these amounts are added.

Comment: Lines185-187: an assumption like this cannot be made; it is too speculative since the phytoplankton community was not sampled at the same time as the shellfish. In any case, it is presumable.

Answer: We appreciate the Referee's comment. Due to the lack of data on phytoplankton community, which was not sampled together with the shellfish, the entire paragraph discussing the phytoplankton has been deleted (Lines 275-282).

Comment: Lines 199-203: DA occurs in which amounts in all these species?

Answer: Maximal concentrations determined in the cited research, as well as the scientific names of the species under research, have been added (Lines 304-309).

Comment: Lines 205-217: for each reference, for each site, specify the amounts of DA measured in which species; otherwise, it is very difficult to contrast the results of this work.

Answer: In Lines 311-321, the range of DA concentrations determined in the research by Ljubešić et a. (2011) and the maximal concentrations determined in the work by Ujević et al. (2010) at each site, as well as the scientific names of the investigated species, have been added.

Comment: Line 218: Table 4 is very difficult to read and understand. Please simplify and modify it with a detailed statistical analysis as suggested in Table 3

Answer: In order to simplify data presentation, Table 4 has been replaced with Figure 2 – a line diagram showing seasonal DA occurrence in three studied species, together with the pertaining standard error.

Comment: Lines 223-231: This section reads more like a summary of results than conclusions. Please rewrite it emphasizing the biological interpretation of the results and the main contributions of the knowledge derived from this work for future research.

Answer: The Conclusion section has been rewritten. Since the Results and the Discussion are united in a single section, due to the extensiveness of this section the summary of the results is given in Lines 357-366. In Lines 366-367, the observation derived from the results of the study is given, followed by the conclusion on the usefulness of this research in planning phycotoxin monitoring in shellfish, in particular when it comes to the selection of suitable sentinels (Lines 368-370). In Lines 370-372, the need for future research that will involve different environmental factors affecting DA accumulation in various species is emphasized, as well as the need for research on DA accumulation in species other than bivalves that could act as vectors to humans.

Comment: Lines 234-235: What other species should be considered as DA-sentinels in monitoring programs? What is the basis for saying this? this statement demerits all the work presented in this paper.

Answer: We agree with the Referee that there is no basis for stating that species other than bivalves should be considered as DA sentinels in monitoring programs. In view of the foregoing, this part of the text has been deleted (Lines 374-377).

Comment: Lines 238-248: specify the coordinates of the sampling site instead of providing oceanographic details about the area. The total shellfish sample quantity in kg does not provide any additional information. Instead, specify the average length (cm) and weight (g) of each species recorded during the annual sampling cycle.

Answer: The sampling site coordinates have been added in Line 381. The datum on the total shellfish sample quantity was removed from the manuscript. Since the differences in accumulation of DA influenced by individual sizes of the species representatives were not a part of this study, the length and the weight of the samples were regrettably not recorded, so that we are unable to provide these data. Samples under this study were composed of individuals of different sizes (spanning from smaller to larger ones, represented in equal proportions).

Comment: Lines 250-295: This section is overwhelmingly long, it needs to be summarized and simplified. The protocol for sample preparation and toxin extraction is very confusing and different from the methods traditionally applied in these kinds of studies. This makes me deeply doubt the feasibility of the results.

Answer: In the paragraph Chemicals and Analytical Standard, CRM FDMT1 was added (lines 392, 393). The rest of the paragraph was replaced with a short information on the purity of solvents and chemicals (Line 401,402). The part of the manuscript in which sample preparation was described, is shortened in accordance with the Referee's comment, so that only the main steps are presented in Lines 404-430. Sample preparation is indeed different from the methods usually used for DA quantification, and much more demanding. The method was chosen due to its capability to detect DA trace levels and is not suitable for routine monitoring. The instrument and the mobile phases used are mentioned in Lines 432-440.

Comment: Lines 296-331: it is unnecessary to give so much detail about the MS parameters and the validation of these analyzes. Please summarize this in a few lines.

Answer: The text originally given in paragraphs dealing with the LC-MS/MS parameters and validation is now summarized in a few sentences providing necessary information on the method SRM transitions used for DA detection (presented in Figure 1). Table 5 and detailed description of MS-MS parameters have been removed. Quality control of the method is described in Lines 454-456.

Comment: Lines 333-334: these lines are incorrectly placed in this section, they should be in the DA-quantification section, and the origin of the DA standard solutions used to construct the calibration curve should be specified.

Answer: We appreciate the Referee's remark. The incorrectly placed lines have now been moved to the LC-MS/MS Analyses section (Line 457). The origin of DA standard solutions used to plot the calibration curve is specified in the Chemicals and Analytical Standard section (Line 395).

Comment: Lines 334-338: More details about the statistical analyzes used in this work are needed, such as what a priori tests were applied to measure the homogeneity of variances and the frequency distribution of the data? What post hoc test was used for the multiple comparisons of means/medians?

Answer: The revised text is supplemented with the information that the normality of data distribution has been checked using the Shapiro-Wilks test, which indicated that the use of non-parametric test is in order (Lines 484, 485). However, the Kruskal-Wallis test, which is a non-parametric version of the One-way ANOVA and lacks the parameters such as variance homogeneity testing and a separate post hoc test, was used, so that the information requested by the Referee was not given in the original text.  

The authors are indebted to our esteemed Reviewer for their most helpful suggestions and comments.

Reviewer 2 Report

Three species of marine organisms were investigated for DA contamination during a yearlong period. More than 200 samples were collected and analyzed in this study. It looks a good and meaningful report for the seasonal variation of DA in the investigative zone. However, there are some comments need to be considered and revised to improve the quality of this manuscript before consideration of accept. 

  1. Most of diagrams need to be improved or completely revised in this manuscript. For example, the accurate LOD and LOQ values in three different sample matrix were not necessary shown in the Table 1. It is enough just to show the average values of LOD and LOQ in the Text because these values are very close and similar in three different samples. The style of chromatograms shown in the Figure 1 was very unsatisfactory because no much information could be got from this figure. Usually the SRM transitions used in this method should be shown and the Table 5 could be omitted.
  2. The style of Table 4 should be changed to line figures in order to show the concentrations of DA in samples with different sampling time. 
  3. The determination method for LOD value should be addressed clearly in the method section. The LOD values obtained in this study was lower than that shown in the reference [41] Beach et al., 2014. What kind of procedures were modified or the instrument used here is more advanced and sensitive? 
  4. A sampling map should be given which can let readers easily get the investigation regions. 
  5. To compare and discuss the accumulation and depuration dynamic process of DA in organisms, the authors should paid attention to the differences between the ecological niche, and the discrepancy of reproductive cycle of these organisms investigated here. 
  6. There are a few formal errors in the text. For example, lines 9 and 33, spp. should be not italic. Line 149, the word "Edulis" should be "edulis".      

Author Response

Comment: Most of diagrams need to be improved or completely revised in this manuscript. For example, the accurate LOD and LOQ values in three different sample matrix were not necessary shown in the Table 1. It is enough just to show the average values of LOD and LOQ in the Text because these values are very close and similar in three different samples. The style of chromatograms shown in the Figure 1 was very unsatisfactory because no much information could be got from this figure. Usually the SRM transitions used in this method should be shown and the Table 5 could be omitted.

Answer: In line with the Referee's comment, we have revised the diagrams contained by our manuscript. Table 1 was removed. Information on LOD and LOQ values is given under the Validation of Analytical Method section in Lines 101-103. LOD and LOQ values are expressed as an approximate value of similar values determined for each species. The chromatogram in Figure 1 was replaced with the figure showing the method SRM transitions used for DA detection. Table 5 has been removed.

Comment: The style of Table 4 should be changed to line figures in order to show the concentrations of DA in samples with different sampling time.

Answer: Table 4 has been replaced with Figure 2 – a line diagram presenting the monthly occurrence of DA in European oysters, Queen scallops and ascidians, witnessed from April 2018 till March 2019. Vertical bars represent the standard error, while various superscripts denote a statistically significant difference.

Comment: The determination method for LOD value should be addressed clearly in the method section. The LOD values obtained in this study was lower than that shown in the reference [41] Beach et al., 2014. What kind of procedures were modified or the instrument used here is more advanced and sensitive?

Answer: The reference stating the methods of LOD and LOQ determination is given in sections 2.1 Validation of Analytical Method (Line 101) and 4.5 Validation (Lines 472-474). LOD vas determined using blanks approach described by Wenzl et al. (2016). Ten blank samples were analysed. LOD was calculated as the ratio of standard deviation of the blank signals over the calibration curve slope (in a lower concentration range) multiplied by 3.9. As for DA quantification, we used the alternative procedure for ultra-trace analysis described by Beach et al., 2014, with the following modifications: a decreased volume for SAX eluate solution (0.2 mL), a decreased total derivatisation volume of 0.3 mL, and a more concentrated borate buffer (150 mM). With the same procedure for ultra-trace analysis, the above authors obtained the LOD value of 1.1 µg/kg. The difference between their and our LOD value could be attributed to the different approach to LOD determination; namely, to determine LOD, Beach et al. (2014) used fortified samples and their signal to noise ratio.

Comment: A sampling map should be given which can let readers easily get the investigation regions.

Answer: The sampling area was described in short under the Materials and Methods section (4.1 The Collection of Samples) (Lines 380-384). We appreciate the Referee's comment on the necessity to pinpoint the study location for the sake of the reading audience. To that effect, the precise study location coordinates have been given (Line 381).

Comment: To compare and discuss the accumulation and depuration dynamic process of DA in organisms, the authors should paid attention to the differences between the ecological niche, and the discrepancy of reproductive cycle of these organisms investigated here.

Answer: In line with the Referee's comment, the literature has been revised so as to improve the quality of this research. The missing aspects have been added (Lines 192 – 247). In Lines 196-210, different kinetic models of depuration in bivalves are described, together with the shellfish species in which DA depuration takes place in some of the above manners. In Lines 216-225, the capability of some species to selectively reject diatoms and a possible effect of DA-utilizing bacteria are mentioned. In Lines 226-238, the influence of feeding pattern on DA accumulation and the existence of diatom protective mechanisms are elaborated. In Lines 238-247, molecular level differences in DA depuration are described. Environmental influences such as temperature, salinity, and food availability, are described, as well (Lines 283-295). The influence of reproductive cycle on DA accumulation is mentioned in Lines 321-329. In the same lines, the lack of research on the reproduction of species investigated in this research and the need for future research tackling this subject are highlighted. The pertaining references are provided, too. The Reference List at the end of the paper is supplemented accordingly.

Comment: There are a few formal errors in the text. For example, lines 9 and 33, spp. should be not italic. Line 149, the word "Edulis" should be "edulis".     

Answer: We appreciate the Referee's remark pointing towards the formal errors in the text. In Line 12, the format in which the word “spp” was written is now corrected. Upon revision of the original text, the line in which the same formal error occurred (Line 38) has been deleted, together with the sentence in which the word “edulis” began with the capital letter (Lines 194-196).

The authors are indebted to our esteemed Reviewer for their most helpful suggestions and comments.

Reviewer 3 Report

  1. Materials and methods: In order to minimize the inter-species differences caused by various environmental conditions, the samples of this study were obtained from the same harvesting area in Northern Adriatic. Detailed sampling stations also need to be described, please provide sampling stations in the form of tables or diagrams.
  2. Validation of Analytical Method: Repeatability, in-house reproducibility, recovery, and measurement uncertainty established for DA determined in the four species were analyzed. A total of 128 spiked samples at four different concentration levels (50, 250, 500, and 1,000 μg/kg) were determined. The recovery range of all spiked samples was higher than 90%, but these results did not represent actual positive samples. It is recommended to use positive tissue reference material for domoic acid when verifying methods.
  3. In this manuscript the authors found that the Queen scallops showed the greatest affinity for DA accumulation than other two species. The authors explained the possible reasons for this result, such as species-specific, elimination rates, environmental conditions. Lack of molecular mechanisms to analyze possible reasons of domoic acid absorption and excretion in shellfish and ascidians.
  4. The authors assumed that DA might be present in shellfish during any season. Seasonal changes may cause significant changes in temperature, salinity and phytoplankton abundance. These may significantly affect the physiological activities of shellfish and ascidians. This hypothesis needed more references to support.
  5. The aim of this study was to determine seasonal differences in occurrence and accumulation of DA in shellfish and ascidians. The author's data analysis was relatively simple. The author compared the changes of same specie with seasons. Is there any analysis of different varieties in the same season? Has the author conducted the risk exposure assessments?

Author Response

Comment: Materials and methods: In order to minimize the inter-species differences caused by various environmental conditions, the samples of this study were obtained from the same harvesting area in Northern Adriatic. Detailed sampling stations also need to be described, please provide sampling stations in the form of tables or diagrams.

Answer: The sampling area was described in short under the Materials and Methods section (4.1 The Collection of Samples) (Lines 380-384). We appreciate the Referee's comment on the necessity to pinpoint the study location for the sake of the reading audience. To that effect, the precise study location coordinates have been given (Line 381).

Comment: Validation of Analytical Method: Repeatability, in-house reproducibility, recovery, and measurement uncertainty established for DA determined in the four species were analyzed. A total of 128 spiked samples at four different concentration levels (50, 250, 500, and 1,000 μg/kg) were determined. The recovery range of all spiked samples was higher than 90%, but these results did not represent actual positive samples. It is recommended to use positive tissue reference material for domoic acid when verifying methods.

Answer: During the validation study, we indeed used positive tissue reference material for domoic acid, that is to say, the NRC CNRCʼs certified reference material containing DA - FDMT1 for recovery check-up during validation (Lines 104-106).

Comment: In this manuscript the authors found that the Queen scallops showed the greatest affinity for DA accumulation than other two species. The authors explained the possible reasons for this result, such as species-specific, elimination rates, environmental conditions. Lack of molecular mechanisms to analyze possible reasons of domoic acid absorption and excretion in shellfish and ascidians.

Answer: In line with the Referee's comment, the literature was rechecked so as to improve the quality of this research. The missing aspects of molecular level DA depuration have been added (Lines 238-247). The pertaining references have been provided, too. The Reference List at the end of the paper has been supplemented accordingly.

Comment: The authors assumed that DA might be present in shellfish during any season. Seasonal changes may cause significant changes in temperature, salinity and phytoplankton abundance. These may significantly affect the physiological activities of shellfish and ascidians. This hypothesis needed more references to support.

Answer: In line with the Referee's comment, the influence of environmental factors such as temperature, salinity, and food availability on DA accumulation has been briefly elaborated (Lines 283-295). Given that these factors also affect seasonal spawn occurrence, which in turn changes shellfish physiological status (and the accumulation of DA, for that matter), we have emphasised the need for future research on yet under-investigated influence of the above factors on spawn (Lines 321-329). The pertaining references have been provided, too. The Reference List at the end of the paper has been supplemented accordingly.

Comment: The aim of this study was to determine seasonal differences in occurrence and accumulation of DA in shellfish and ascidians. The author's data analysis was relatively simple. The author compared the changes of same specie with seasons. Is there any analysis of different varieties in the same season? Did the author conducted the risk exposure assessments?

Answer: The statistics was originally provided for each study month, not a season (i.e., a few months), because the authors wanted to investigate into the monthly rather than seasonal variability. However, in line with the Referee’s comment, the standard error descriptive of each month has been calculated and made a part of the descriptive statistics. Given that one of our esteemed Referees requested a line diagram display, the standard error has been given within that diagram (Figure 2). Since no risk assessment was foreseen by the original study design, risk exposure analyses were not performed.

The authors are indebted to our esteemed Reviewer for their most helpful suggestions and comments.

Round 2

Reviewer 1 Report

It is impossible to review the new version of this manuscript under these conditions. A messy document that is extremely difficult to read.

Author Response

The authors are indebted to our esteemed Reviewers for their most helpful suggestions and comments, to which we made every effort to fully respond. The revised manuscript has been rechecked by a highly qualified linguist (University Professor) having a vast experience in scientific writing and MS reviewing.

Reviewer 2 Report

The original version of this ms has been completely revised referring to the comments of reviewers. Its quality has been significantly improved now. However, I think the novelty was not clearly shown in this paper. But I think it could be considered as a publication in the type of Short communication.  

Author Response

The authors are indebted to our esteemed Reviewers for their most helpful suggestions and comments, to which we made every effort to fully respond.